# ROYAL SOCIETY
# OPEN SCIENCE

environmental science/spectroscopy

laser-induced fluorescence, fluorescence spectrum, first-derivative fluorescence, leaf nitrogen concentration

**Author for correspondence:**
Jian Yang
e-mail: yangjian@cug.edu.cn

This article has been edited by the Royal Society of Chemistry, including the commissioning, peer review process and editorial aspects up to the point of acceptance.

# Estimating leaf nitrogen concentration based on the combination with fluorescence spectrum and first-derivative

Jian Yang[1], Lin Du[2], Wei Gong[3], Shuo Shi[3] and Jia Sun[2]

[1]Artificial Intelligence School, Wuchang University of Technology, Wuhan, Hubei 430223, People's Republic of China
[2]School of Geography and Information Engineering, China University of Geosciences, Wuhan, Hubei 430074, People's Republic of China
[3]State Key Laboratory of Information Engineering in Surveying, Mapping and Remote Sensing, Wuhan University, Wuhan, Hubei 430072, People's Republic of China

JY, 0000-0003-3621-4169

Leaf nitrogen concentration (LNC) is a major indicator in the estimation of the crop growth status which has been diffusely applied in remote sensing. Thus, it is important to accurately obtain LNC by using passive or active technology. Laser-induced fluorescence can be applied to monitor LNC in crops through analysing the changing of fluorescence spectral information. Thus, the performance of fluorescence spectrum (FS) and first-derivative fluorescence spectrum (FDFS) for paddy rice (Yangliangyou 6 and Manly Indica) LNC estimation was discussed, and then the proposed FS + FDFS was used to monitor LNC by multivariate analysis. The results showed that the difference between FS ($R^2 = 0.781$, s.d. = 0.078) and FDFS ($R^2 = 0.779$, s.d. = 0.097) for LNC estimation by using the artificial neural network is not obvious. The proposed FS + FDFS can improved the accuracy of LNC estimation to some extent ($R^2 = 0.813$, s.d. = 0.051). Then, principal component analysis was used in FS and FDFS, and extracted the main fluorescence characteristics. The results indicated that the proposed FS + FDFS exhibited higher robustness and stability for LNC estimation ($R^2 = 0.851$, s.d. = 0.032) than that only using FS ($R^2 = 0.815$, s.d. = 0.059) or FDFS ($R^2 = 0.801$, s.d. = 0.065).

## 1. Introduction

All organic compounds were supplied by the photosynthesis of crops, thus it is of critical significance to accurately estimate crops' physiological conditions based on different methods. Leaf biochemical parameters served as the vital indictors for estimating

crops' growth status both locally and globally, and have been widely used in many fields. Leaf nitrogen concentration (LNC) is a key parameter of vegetation photosynthetic efficiency and yields, which can be applied for scientific guidance in nitrogen (N) fertilization management [1–3]. Hence, fast and accurate estimation of crops' LNC is vital to indirectly understand crops' growth status [4–6]. The traditional chemical analysis methods can accurately obtain the value of LNC, but it is time-consuming and costly, which restricted its application in the fast and large scale. Remote sensing technology serves as an alternative method, which relies on the interaction of light and leaf medium, and can be applied in the estimation of crop physiological condition [7–9].

Based on the passive hyperspectral remote sensing technology, studies have been conducted by some scholars to build different models to obtain more accurate biochemical parameter estimation. They proposed different vegetation indices based on statistical analysis models for estimating biochemical parameters or crops yields [10–13]. However, passive hyperspectral technology is difficult to acquire the spatial information. Thus, the combination of passive spectrum and lidar was proposed to monitor the growth status of vegetation [14,15]. In addition, the hyperspectral lidar technology was also proposed and applied in vegetation monitoring [16–18]. These methods or technologies can efficiently improve the accuracy of vegetation biochemical parameter estimation by affording more characteristic information.

Compared with the reflectance spectrum, chlorophyll fluorescence serving as an alternative means has been diffusely used in the estimation of vegetation growth status with the properties of being non-destructive, rapid and highly sensitive [19–21]. Chlorophyll fluorescence is generated when chlorophyll is irradiated with incident light of a certain wavelength and then radiates part of its absorbed energy at longer wavelengths. At present, the fluorescence technology has been used in vegetation monitoring including laser-induced fluorescence transient (LIFT) [22], fast fluorescence kinetics [23] and laser-induced fluorescence (LIF) [24]. LIFT can acquire the changing of fluorescence signals at certain wavelength with time, which can be used to estimate the biochemical concentration of crops. Fast fluorescence kinetics can detect the changing of fluorescence signals with time. Fast fluorescence kinetics were usually applied in the monitoring of photosystem [12,25]. LIF spectra can be measured by using a laser to excite, which included amounts of spectral information. However, LIFT and fast fluorescence kinetics are related with time, which restricted their large-scale application [26]. Thus, fluorescence spectral information was applied more widely in remote sensing [9,27].

McMurtrey et al. [28] compared the ability of the fluorescence and reflectance to measure different nitrogen (N) levels of plant stress in corn crops. Then, they found that the fluorescence ratio F740/F685 or F525/F685 can serve as early indication of metabolic stress. In addition, Gunther et al. found that F685/F730 is only correlated to the chlorophyll content [29]. Subhash & Mohanan [30] discussed the ability of the chlorophyll fluorescence (650–800 nm) and pointed out that red fluorescence spectrum (FS) has great potential for nutrition stress monitoring in rice. Gameiro et al. [31] used the LIF means to analyse water stress in Arabidopsis and found that LIF can be applied in precocious detection. Yang et al. [32,33] analysed the ability of LIF parameters with the help of multivariate analysis to estimate LNC. In addition, the combination with fluorescence parameters and reflectance spectrum has also been analysed in some applications [34–38]. In order to obtain more fluorescence characteristics, the first-derivative fluorescence spectrum (FDFS) was proposed and used to monitor LNC [39].

Nevertheless, comparative studies on the abilities of FS and FDFS to monitor LNC are still limited. What is more, there is no relative literature conducted on studying the performance of the combination of FS and FDFS for LNC estimation. Thus, the main purpose of this study is to: (i) compare the abilities of FS and FDFS to estimate LNC based on multivariate analysis, and (ii) discuss whether FS combined with FDFS can improve the accuracy of paddy rice LNC estimation.

# 2. Materials and experiment

## 2.1. Samples

Yangliangyou 6 was cultivated in 2015, and Manly Indica was cultivated in 2016. The experimental area is sited in the Jianghan China Plain and the latitude and longitude ranges from 29°58′ N to 31°22′ N and 113°41′ E to 115°05′ E, respectively. In order to obtain different LNCs, different fertilization levels were used in the experimental areas. Different N fertilization levels of urea were used in 2015 (0, 120, 180 and 240 kg ha$^{-1}$) and 2016 (0 and 150 kg ha$^{-1}$). According to the suggestion of the local farm

extension service, 180 and 150 kg ha$^{-1}$ were the optimal N fertilization in 2015 and 2016, respectively. Then, an absolute block design, in which ridges of the field were enclosed in plastic films to avoid water leakage, was used for each experimental area to maintain the same cultivation surroundings. In this study, the samples in this study were selected from the second leaves above. Foliar samples were fully unfolded and gathered by randomly collecting nine samples with three replicates for each experimental field on 26 July 2015 and 23 July 2016, respectively. Thus, the total number of foliar samples is 486. The gathering period of rice is tillering stage.

## 2.2. Measurement of fluorescence spectra

LIF equipment was set up in our laboratory and was used to collect the fluorescence spectrum of leaves [33]. A neodymium-doped yttrium aluminium garnet laser, which emitted 1064 nm seeds light and then obtained 355 nm excitation light through a third harmonic generation, served as the excitation light source in this study. To decrease the effect of background fluorescence on the chlorophyll fluorescence signals, all foliar samples were fixed on black cardboard which was kind of non-fluorescent material [40]. The excitation light was irradiated perpendicularly on the surface of the foliar samples, and the emission fluorescence signals were detected at the backward direction of the same side. The spectral range was between 360 and 800 nm which can include the chlorophyll fluorescence spectral region 650–800 nm. The spectral resolution was 0.5 nm. After the fluorescence spectrum measurement was completed, these foliar samples were sent to measure biochemical concentration. In this study, the LNC was determined by using Kjeldahl method [4]. The detailed description of Kjeldahl can be found in [41].

## 2.3. Theory

Based on the previous studies and concepts of first-derivative [42,43], FDFS ($I^{'}(\lambda_i)$) at $\lambda_i$ is the difference between the fluorescence intensity at each band, minus and plus one wavelength, divided by the range of wavelength, as written:

$$I'(\lambda_i, \lambda_{ex}) = \frac{I(\lambda_{i+1}, \lambda_{ex}) - I(\lambda_{i-1}, \lambda_{ex})}{\lambda_{i+1} - \lambda_{i-1}}, \tag{2.1}$$

where $\lambda_{ex}$ is the wavelength of excitation light, $I(\lambda_{i+1}, \lambda_{ex})$ and $I(\lambda_{i-1}, \lambda_{ex})$ denote the excited fluorescence signals at $\lambda_{i+1}$ and $\lambda_{i-1}$, respectively; $i$ is the corresponding wavelength.

## 2.4. Analytical methods

Back-propagation neural network (BPNN) can be used in analysing multifarious nonlinear problems by establishing the relationship between the inner neural units based on a succession of trials [44]. This algorithm has been widely used in many fields due to its abilities of self-adaption and self-learning [45]. In this study, a three-layer neural network was selected to analyse the LNC based on the built inverse model. According to previous research [11], the 'trainbr' was used to serve as the training function. In addition, the size of hidden layer was set as four in this model.

Principal component analysis (PCA) can efficiently reduce the number of original parameters by extracting the key characteristics variables and deleting lower-level components. However, the major spectral information will not be lost [46]. Thus, PCA served as a statistical multivariate analysis method and was applied in data dimension reduction [47]. The extracted variables can be calculated by the linear combinations of the original variables. Therefore, the analysis procedure will be greatly simplified by using fewer extracted parameters than the original data [48].

The fluorescence features were stochastically divided into two datasets: 70% as the training dataset and 30% as the validation set. In this study, coefficient of determination ($R^2$) and standard deviation (s.d.) were used to evaluate the capability of the BPNN by using different feature variables (fluorescence, FDFS, combination of fluorescence and FDFS). The s.d. was calculated as follows:

$$\text{s.d.} = \sqrt{\frac{1}{n}\sum_{i=1}^{n}(X_i - u)^2}, \tag{2.2}$$

where $X_i$ denotes the value of each $R^2$ based on BPNN model; and $n$ represents the repeated times. Low s.d. and high $R^2$ denote an excellent capability of training model. To analyse the stability of the

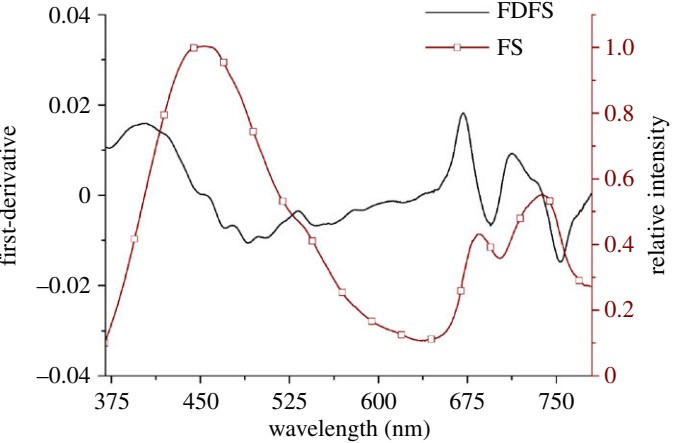

**Figure 1.** Fluorescence spectrum and FDFS of paddy rice leaf with 355 nm excitation light.

fluorescence characteristics for LNC estimation, each feature setting was repeated 1000 times by using BPNN model. Then, the average and s.d. of $R^2$ can be obtained. In this paper, Matlab R2015b (Mathworks Inc., Natick, MA, USA), where a lot of regression toolboxes are available, was used to analyse all data.

# 3. Results and discussion

## 3.1. First-derivation spectrum

FS excited by 355 nm excitation light and corresponding to the FDFS value of each band obtained by using equation (2.1) are shown in figure 1.

Figure 1 demonstrated that the FDFS showed more fluorescence characteristics than the fluorescence spectrum. The fluorescence spectrum excited by 355 nm displayed three main characteristics peaks at 440–460 nm, 680–690 nm and 735–745 nm. The first fluorescence characteristic peak was mainly attributed to the nicotinamide adenine dinucleotide [49,50]. The fluorescence characteristics peaks at 735–745 nm, and 680–690 nm correspond to antenna chlorophyll of Photosystems I and II and Chlorophyll a of Photosystem II, respectively [51]. The FDFS exhibited a shoulder peak at 720 nm; three fluorescence peaks at 530 nm, 675 nm and 700 nm; and three valleys at 485 nm, 690 nm and 750 nm. The values of FDFS transferred from negative to positive is the fluorescence spectral valley, which is generally ignored in LNC inversion. Compared with fluorescence spectrum, FDFS can afford additional fluorescence features for LNC estimation.

## 3.2. LNC estimation by using spectral information

To discuss the performance of the spectral information for LNC estimation, then BPNN model was used to inverse LNC according to the different fluorescence spectral characteristics in this study. LNC and different fluorescence spectral features corresponded, respectively, to the output and input parameters in the BPNN. In order to analyse the robustness of fluorescence features, each characteristic setting was repeated 1000 times by using BPNN. The distribution of $R^2$-values between the measured and predicted LNC is shown in figure 2.

Figure 2 shows the distribution of $R^2$-values based on BPNN model for different fluorescence spectral features. It is clear that the performance of FS + FDFS is superior to that of only FS or FDFS for LNC estimation. The results demonstrated that the FDFS can provide more fluorescence features to improve the accuracy of LNC estimation. In addition, the variation range of $R^2$-values is listed in table 1.

From table 1, the difference of the maximum and minimum of $R^2$ is not obvious for different fluorescence spectral features. However, the robustness of FS + FDFS (s.d. = 0.051) is superior to that using only FS (s.d. = 0.078) or FDFS (s.d. = 0.097) for LNC estimation. Thus, the proposed FS features combined with FDFS can be applied to improve the accuracy of the monitoring of crops' growth status to a certain extent without additional measurements. In addition, we can find that this improvement is not very obvious. The main reason is that the FS and FDFS are information redundant and highly

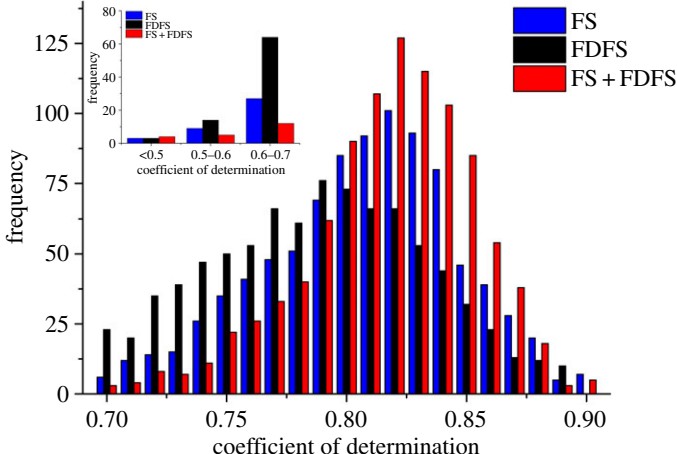

**Figure 2.** $R^2$ histogram distribution for different fluorescence spectral characteristics by using BPNN model. The small graph is the $R^2$ histogram distribution which is lower than 0.7.

**Table 1.** The performance of fluorescence spectra and first-derivative fluorescence spectra by using BPNN model.

| | $R^2$ | | | |
|---|---|---|---|---|
| | min | max | mean | s.d. |
| FS | 0.480 | 0.936 | 0.781 | 0.078 |
| FDFS | 0.352 | 0.925 | 0.779 | 0.097 |
| FS + FDFS | 0.478 | 0.941 | 0.813 | 0.051 |

autocorrelated between the adjacent wavelengths with 0.5 nm sample interval, which may impact on the its ability for the LNC monitoring. Thus, PCA was applied in this study to reduce dimensionality FS and FDFS, and then extract main fluorescence characteristics for LNC estimation.

## 3.3. LNC estimation by using PCA

PCA was used to analyse FS and FDFS, then the extracted main fluorescence features served as input variables in this model to train BPNN to improve accuracy of LNC estimation. In order to understand the extracted characteristics variables by applying PCA, the loading weight of the first three principal components (PCs) is given in figure 3, which includes over 90% explained variables.

Figure 3 shows that the main fluorescence characteristics peaks can be included by the extracted characteristics variables based on PCA. Then, the calculated new fluorescence variables served as input variables to train BPNN to estimate LNC. Then, the distribution of $R^2$-values can be obtained by using the BPNN model with 1000 repeats for each setting data (figure 4).

Figure 4 is the histogram distribution of $R^2$ based on BPNN model for different fluorescence spectral features, which was processed by using PCA. We can find that the performance of the extracted variables is better than the original spectral features (figure 2) for LNC estimation. The results demonstrated that PCA can be efficiently applied in the analysis of spectral information to extract the main characteristics variables and reduce the information redundancy. In addition, figure 4 shows that the calculated new variables based on FS + FDFS exhibited better performance for LNC estimation than that only using the calculated new variables based on FS or FDFS. To better comprehend the performance of the calculated new variables for LNC estimation, the variation range of $R^2$-values is listed in table 2.

Table 2 displays the difference of the maximum and minimum of $R^2$-values for LNC estimation based on different extracted features variables by using BPNN model. By comparing the mean of $R^2$-values and s.d., it can be found that FS ($R^2 = 0.815$, s.d. $= 0.059$) is superior to FDFS ($R^2 = 0.805$, s.d. $= 0.065$) for LNC monitoring by using the PCA combined with BPNN algorithm. The possible reason is that FDFS is sensitive to oscillations and will increase the risk of possible scattering of the data. Although the smoothing of data has been processed, it will still influence the accuracy of LNC monitoring. The

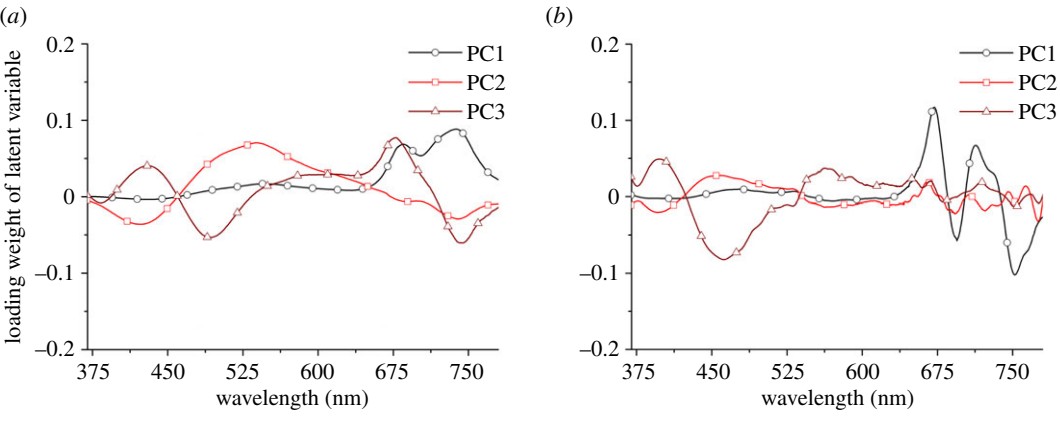

**Figure 3.** The loading weights of the first three principal components. (*a*) Fluorescence spectra; (*b*) first-derivative fluorescence spectra.

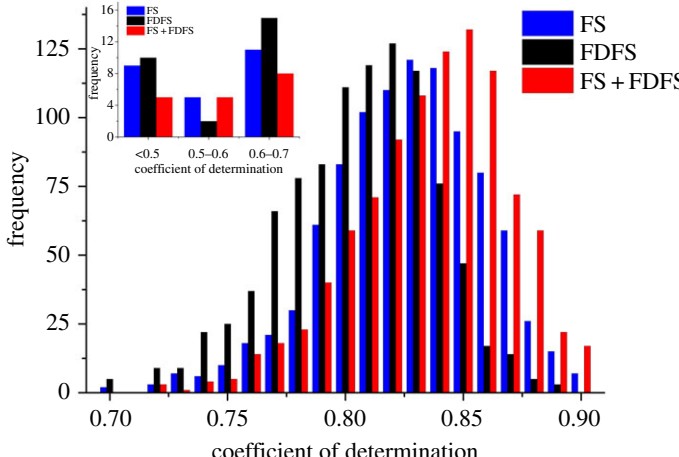

**Figure 4.** $R^2$ histogram distribution for different fluorescence characteristics with PCA combined with BPNN model. The small graph is the $R^2$ histogram distribution which is lower than 0.7.

**Table 2.** The performance of fluorescence spectra and first-derivative fluorescence spectra by using the combination of PCA and BPNN model.

|  | $R^2$ | | | |
|---|---|---|---|---|
|  | min | max | mean | s.d. |
| FS | 0.455 | 0.914 | 0.815 | 0.059 |
| FDFS | 0.256 | 0.901 | 0.801 | 0.065 |
| FS + FDFS | 0.542 | 0.921 | 0.851 | 0.032 |

detailed analysis needs to be further analysed in the following work. Compared to using only FS or FDFS, the FS + FDFS displayed better stability for LNC monitoring with higher $R^2$ mean values and lower s.d. The satisfactory results based on the FS + FDFS demonstrated its application potential for the LNC monitoring.

The performance of FD and FDFS for LNC monitoring by using PCA combined with BPNN was discussed in this study, and then FS + FDFS was proposed to estimate LNC. The combination with FS and FDFS displayed application potential for LNC estimation. However, the effect of the oscillation of FDFS on the accuracy of LNC monitoring still needs to be further discussed. In addition, the effect of multivariate analysis on accuracy also needs be considered in the following work.

# 4. Conclusion

In this study, the predictive ability of FS and FDFS for LNC estimation was discussed, and then the combination with FS and FDFS was proposed to monitor LNC by using the PCA and BPNN algorithms. The experimental results indicated that the difference between FS ($R^2 = 0.781$, s.d. = 0.078) and FDFS ($R^2 = 0.779$, s.d. = 0.097) for LNC estimation by applying the BPNN is not obvious. The proposed FS + FDFS can improve the accuracy and stability of LNC estimation to some extent ($R^2 = 0.813$, s.d. = 0.051). Then, PCA was used to analyse FS and FDFS and to reduce the verbose information. The proposed FS + FDFS exhibited higher robustness for LNC estimation ($R^2 = 0.851$, s.d. = 0.032) than that using only FS ($R^2 = 0.815$, s.d. = 0.059) or FDFS ($R^2 = 0.801$, s.d. = 0.065) based on the PCA combined with BPNN model. Therefore, the study results demonstrated that FS combined with FDFS can improve the accuracy and robustness of LNC monitoring based on multivariate analysis without any additional measurements.

Data accessibility. Data available from the Dryad Digital Repository: https://doi.org/10.5061/dryad.pg4f4qrjc [52].

Authors' contributions. J.Y., L.D., W.G., S.S. and J.S. conceived the study and contributed to the dataset. J.Y. analysed the data and wrote the manuscript.

Competing interests. We declare we have no competing interests.

Funding. This research was funded by the National Key Research and Development Program of China (grant no. 2018YFB0504500), the National Natural Science Foundation of China (grant no. 41801268) and Fundamental Research Funds for the Central Universities, China University of Geosciences (Wuhan) (grant no. CUG170661).

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
