## [Reviewer comments · Royal Society Open Science]

Review History

RSOS-191941.R0 (Original submission)

Review form: Reviewer 1

Is the manuscript scientifically sound in its present form?

Yes

Are the interpretations and conclusions justified by the results?

Yes

Is the language acceptable?

Yes

Do you have any ethical concerns with this paper?

No

Have you any concerns about statistical analyses in this paper?

No

Recommendation?

Accept as is

Comments to the Author(s)

Please see the attached file (Appendix A).

Review form: Reviewer 2

Is the manuscript scientifically sound in its present form?

Yes

Are the interpretations and conclusions justified by the results?

Yes

Is the language acceptable?

Yes

Do you have any ethical concerns with this paper?

No

Have you any concerns about statistical analyses in this paper?

No

Recommendation?

Accept with minor revision (please list in comments)

Comments to the Author(s)

The manuscript presents interesting results on application of fluorescence excitation spectra for estimation of leaf nitrogen content in rice leaves. The innovativeness of the study is in application of first-derivative spectra instead of traditional approaches, which led to improvements in estimation when using the multivariate analysis.

The manuscript is very well written, the text is clear and concise. The abstract is mostly informative and accurate. The study is well introduced, including clear aims. The description of material and methods is very brief, but mostly sufficient. Results are well presented and appropriately discussed. Overall, the quality of the manuscript is high, and the results are worth to be published. I recommend accepting the manuscript for publication after addressing my comments within MINOR REVISION.

Comments:

1. Based on the title or abstract, the reader has no idea what kind of crop the subject of the study was. Please, provide the information into the abstract, including the type of rice and genotypes used.
2. Please, explain more in detail the "an absolute block design" used in this study.
3. Please, indicate what kind of mathematical or statistical software was used for calculations and analyses.
- 4 In Fig. 2 and 4 captions, put the information about the small graphs (inserts) added together with the main distribution graphs.

Review form: Reviewer 3

Is the manuscript scientifically sound in its present form?

Yes

Are the interpretations and conclusions justified by the results?

Yes

Is the language acceptable?

Yes

Do you have any ethical concerns with this paper?

No

Have you any concerns about statistical analyses in this paper?

No

Recommendation?

Accept with minor revision (please list in comments)

Comments to the Author(s)

The authors present a method useful for non-invasive estimation of paddy rice leaf nitrogen concentration, which has been widely utilized in remote sensing. In this study, the laser-induced fluorescence (LIF) was analyzed using the fluorescence spectrum (FS) and first-derivative fluorescence spectrum (FDFS). And the principal component analysis was used for main fluorescence characteristics extraction. The proposed method is quite innovative, and, based on the results, seems to be quite efficient.

1. Authors are advised to show more details on the materials, such as the growing period, picking time, and the fluorescence lidar detection time.
2. Any experimental photo is preferred.
3. Why PCA is selected and its principle should be added.
4. A space is generally placed between the unit of measure and the quantity. Percentages and angles are exempted from this rule.
5. The font size in figures should be resized to be clear.

Decision letter (RSOS-191941.R0)

17-Jan-2020

Dear Dr Yang:

Title: Estimating leaf nitrogen concentration based on the combination with fluorescence spectrum and first-derivative
Manuscript ID: RSOS-191941

Thank you for submitting the above manuscript to Royal Society Open Science. On behalf of the Editors and the Royal Society of Chemistry, I am pleased to inform you that your manuscript will be accepted for publication in Royal Society Open Science subject to minor revision in accordance with the referee suggestions. Please find the reviewers' comments at the end of this email.

The reviewers and handling editors have recommended publication, but also suggest some minor revisions to your manuscript. Therefore, I invite you to respond to the comments and revise your manuscript.

Because the schedule for publication is very tight, it is a condition of publication that you submit the revised version of your manuscript before 26-Jan-2020. Please note that the revision deadline

will expire at 00.00am on this date. If you do not think you will be able to meet this date please let me know immediately.

Best wishes,
Dr Laura Smith
Publishing Editor, Journals

RSC Associate Editor:
Comments to the Author:
(There are no comments.)

RSC Subject Editor:
Comments to the Author:
(There are no comments.)

Reviewer comments to Author:
Reviewer: 1

Comments to the Author(s)
Please see the attached file

Reviewer: 2

Comments to the Author(s)
The manuscript presents interesting results on application of fluorescence excitation spectra for estimation of leaf nitrogen content in rice leaves. The innovativeness of the study is in application of first-derivative spectra instead of traditional approaches, which led to improvements in estimation when using the multivariate analysis.
The manuscript is very well written, the text is clear and concise. The abstract is mostly informative and accurate. The study is well introduced, including clear aims. The description of material and methods is very brief, but mostly sufficient. Results are well presented and appropriately discussed. Overall, the quality of the manuscript is high, and the results are worth to be published. I recommend accepting the manuscript for publication after addressing my comments within MINOR REVISION.

Comments:

1. Based on the title or abstract, the reader has no idea what kind of crop the subject of the study was. Please, provide the information into the abstract, including the type of rice and genotypes used.
2. Please, explain more in detail the “an absolute block design” used in this study.
3. Please, indicate what kind of mathematical or statistical software was used for calculations and analyses.
- 4 In Fig. 2 and 4 captions, put the information about the small graphs (inserts) added together with the main distribution graphs.

Reviewer: 3

Comments to the Author(s)
The authors present a method useful for non-invasive estimation of paddy rice leaf nitrogen concentration, which has been widely utilized in remote sensing. In this study, the laser-induced fluorescence (LIF) was analyzed using the fluorescence spectrum (FS) and first-derivative fluorescence spectrum (FDFS). And the principal component analysis was used for main fluorescence characteristics extraction. The proposed method is quite innovative, and, based on the results, seems to be quite efficient.

1. Authors are advised to show more details on the materials, such as the growing period, picking time, and the fluorescence lidar detection time.
2. Any experimental photo is preferred.

3. Why PCA is selected and its principle should be added.
4. A space is generally placed between the unit of measure and the quantity. Percentages and angles are exempted from this rule.
5. The font size in figures should be resized to be clear.

Author's Response to Decision Letter for (RSOS-191941.R0)

See Appendix B.

Decision letter (RSOS-191941.R1)

30-Jan-2020

Dear Dr Yang:

Title: Estimating leaf nitrogen concentration based on the combination with fluorescence spectrum and first-derivative

Manuscript ID: RSOS-191941.R1

It is a pleasure to accept your manuscript in its current form for publication in Royal Society Open Science. The chemistry content of Royal Society Open Science is published in collaboration with the Royal Society of Chemistry.

RSC Associate Editor
Comments to the Author:
(There are no comments.)

Reviewer(s)' Comments to Author:

Appendix A

Authors utilized the ANN to estimate the LNC, with the data of FS and FDFS. And then the PCA was used for extraction of the main fluorescence characteristics. The experimental design is sound and the experimental result is Inspiring. This manuscript is well written and should be considered for publication. To improve the quality of this paper, I have several concerns.

1. Presentation Quality: the authors' English is acceptable but still needs a little improvement.
2. Page 4, line 35. The ξ should be in the form of mathematics formula.
3. What is the meaning of the i in eq(1)?
4. Why second-derivative FS is not used for estimation?
5. The use of PCA should be showed in section3.
6. The font size in figures 2 and 3 should be larger.

Appendix B

Dr. Jian Yang
Artificial Intelligence School, Wuchang
University of Technology, Wuhan, 430223,
Hubei, China
Email: yangjian@cug.edu.cn

Manuscript ID: RSOS-191941 to *Royal Society Open Science*

Title: Estimating leaf nitrogen concentration based on the combination with fluorescence spectrum and first-derivative

Author: Jian Yang ^{1*}; Lin Du ²; Wei Gong ³; Shuo Shi ³; Jia Sun ²

Dear Editor-in-chief,

Thanks for the valuable comments from you and the reviewers. We have revised our paper very carefully under those comments and suggestions that helped us a lot to improve the quality of this paper.

Below are our responses to the comments:

Reviewer comments to Author:

Reviewer: 1

Authors utilized the ANN to estimate the LNC, with the data of FS and FDFS. And then the PCA was used for extraction of the main fluorescence characteristics. The experimental design is sound and the experimental result is Inspiring. This manuscript is well written and should be considered for publication. To improve the quality of this paper, I have several concerns.

1. Presentation Quality: the authors' English is acceptable but still needs a little improvement.

Response:

Thank you very much for your comments. According to the advices, the manuscripts has been checked again.

2. Page 4, line 35. The Xi should be in the form of mathematics formula.

Response:

Thank you very much for your helpful advices.

“Xi” has been revised to the form of mathematics formula “ X_i ”.

3. What is the meaning of the i in eq(1)?

Response:

Thank you very much for your helpful advices.

“ i ” denotes the corresponding wavelength in this manuscript. Which relative content has been added “ i is the corresponding wavelength”.

4. Why second-derivative FS is not used for estimation?

Response:

Thank you very much for your helpful advices.

In this paper, the major target is to analyze the performance of fluorescence spectrum (FS) and first-derivative fluorescence spectrum (FDFS) for paddy rice (Yangliangyou 6 and Manly Indica) LNC estimation, and then the proposed FS+FDFS was used to monitor LNC by multivariate analysis. Thus, the second-derivative FS was not used in this paper. However, the application of second-derivative FS in the LNC estimation is good ideas, we will discuss in the next work.

5. The use of PCA should be showed in section3.

Response:

Thank you very much for your helpful advices.

The description of PCA was showed in 3.4. Analytical Methods “Principal Component Analysis (PCA) can efficiently reduce the number of original parameters by extracting the key characteristics variables and deleting lower-level components. However, the major spectral information will be not lost [47]. Thus, PCA served as a statistical multivariate analysis method was applied in data dimension reduction [48].

The extracted variables can be calculated by the linear combinations of the original variables. Therefore, the analysis procedure will be greatly simplified by utilizing fewer extracted parameters than the original data [49].”

6. The font size in figures 2 and 3 should be larger.

Response:

Thank you very much for your helpful advices.

The figures were prepared according to the Author Instruction, which with 300dpi. Thank you very much for your help.

Reviewer: 2

Comments to the Author(s)

The manuscript presents interesting results on application of fluorescence excitation spectra for estimation of leaf nitrogen content in rice leaves. The innovativeness of the study is in application of first-derivative spectra instead of traditional approaches, which led to improvements in estimation when using the multivariate analysis.

The manuscript is very well written, the text is clear and concise. The abstract is mostly informative and accurate. The study is well introduced, including clear aims. The description of material and methods is very brief, but mostly sufficient. Results are well presented and appropriately discussed. Overall, the quality of the manuscript is high, and the results are worth to be published. I recommend accepting the manuscript for publication after addressing my comments within MINOR REVISION.

Comments:

1. Based on the title or abstract, the reader has no idea what kind of crop the subject of the study was. Please, provide the information into the abstract, including the type of rice and genotypes used.

Response:

Thank you very much for your helpful advices.

According to your advices, the relative has been added in abstract “.....first-derivative fluorescence spectrum (FDFS) for paddy rice (Yangliangyou 6 and Manly Indica) LNC estimation was discussed.....”

2. Please, explain more in detail the “an absolute block design” used in this study.

Response:

Thank you very much for your helpful advices.

The absolute block design is mean that ridges of the paddy field were enclosed in plastic films to avoid water leakage [1].

According to your advices, the relative content has been added in this paper. “.....an absolute block design, which ridges of the field were enclosed in plastic films to avoid water leakage, was utilized.....”

Reference:

[1] Song S., Gong W., Zhu B., et al. Wavelength selection and spectral discrimination for paddy rice, with laboratory measurements of hyperspectral leaf reflectance[J]. ISPRS Journal of Photogrammetry & Remote Sensing, 2011, 66(5):672-682.

3. Please, indicate what kind of mathematical or statistical software was used for calculations and analyses.

Response:

Thank you very much for your helpful advices.

All calculation based on the MATLAB software.

According to your advices, the relative content has been added “In this paper, MATLAB R2015b (Mathworks Inc., Natick, MA, USA), where a lot of regression toolboxes are available, was used to analyze all data.”.

4 In Fig. 2 and 4 captions, put the information about the small graphs (inserts) added together with the main distribution graphs.

Response:

Thank you very much for your helpful advices.

According to your advices, the relative content has been added “Figure 2. R^2 histogram distribution for different fluorescence spectral characteristics by using BPNN model. The small graph is the R^2 histogram distribution which is lower 0.7”. “Figure 4. R^2 histogram distribution for different fluorescence characteristics with PCA combined with BPNN model. The small graph is the R^2 histogram distribution which is lower 0.7”.

Reviewer: 3

Comments to the Author(s)

The authors present a method useful for non-invasive estimation of paddy rice leaf nitrogen concentration, which has been widely utilized in remote sensing. In this study, the laser-induced fluorescence (LIF) was analyzed using the fluorescence spectrum (FS) and first-derivative fluorescence spectrum (FDFS). And the principal component analysis was used for main fluorescence characteristics extraction. The proposed method is quite innovative, and, based on the results, seems to be quite efficient.

1. Authors are advised to show more details on the materials, such as the growing period, picking time, and the fluorescence lidar detection time.

Response:

Thank you very much for your helpful advices.

According to your advices, the relative content has been added in 3.1. Samples “Foliar samples were fully unfolded and gathered by randomly collecting 9 samples with three replicates for each experimental field on July 26, 2015 and July 23, 2016, respectively. Thus, the total foliar samples number are 486. The gathering period of rice

is tillering stage.”; “After the fluorescence spectrum measurement was completed, these foliar samples were sent to measure biochemical concentration. In this study, the LNC was determined by using Kjeldahl method.”

2. Any experimental photo is preferred.

Response:

Thank you very much for your helpful advices.

The experimental equipment was set up according to the references [34], the detailed information can be found in Ref. [34]. Thus, the experimental photo was not repeat in this paper.

3. Why PCA is selected and its principle should be added.

Response:

Thank you very much for your helpful advices.

PCA can efficiently reduce the number of original parameters by extracting the key characteristics variables and deleting lower-level components. In addition, the major spectral information will be not lost. Thus, PCA was selected in this paper. In addition, the first three PCs have been included over 90% original information. Thus, the PCs was not added in this paper.

The relative content has been added “.....was given in Fig. 3, which included over 90% explained variables.”

4. A space is generally placed between the unit of measure and the quantity. Percentages and angles are exempted from this rule.

Response:

Thank you very much for your helpful advices.

According to your advices, a space between the unit of measure and quantity has been added in this paper.

5. The font size in figures should be resized to be clear.

Response:

Thank you very much for your helpful advices.

The figures were prepared according to the Author Instruction, which with 300dpi.

Thank you very much for your help.